# Generating giant and tunable nonlinearity in a macroscopic mechanical resonator from a single chemical bond

Pu Huang[1,2,3,*], Jingwei Zhou[1,2,*], Liang Zhang[1,2,*], Dong Hou[1,2,3], Shaochun Lin[1,2,3], Wen Deng[1,2,4], Chao Meng[1,2], Changkui Duan[1,2], Chenyong Ju[1,2,3], Xiao Zheng[1,2,3], Fei Xue[4] & Jiangfeng Du[1,2,3]

Nonlinearity in macroscopic mechanical systems may lead to abundant phenomena for fundamental studies and potential applications. However, it is difficult to generate nonlinearity due to the fact that macroscopic mechanical systems follow Hooke's law and respond linearly to external force, unless strong drive is used. Here we propose and experimentally realize high cubic nonlinear response in a macroscopic mechanical system by exploring the anharmonicity in chemical bonding interactions. We demonstrate the high tunability of nonlinear response by precisely controlling the chemical bonding interaction, and realize, at the single-bond limit, a cubic elastic constant of $1 \times 10^{20}\,\mathrm{N\,m^{-3}}$. This enables us to observe the resonator's vibrational bi-states transitions driven by the weak Brownian thermal noise at 6 K. This method can be flexibly applied to a variety of mechanical systems to improve nonlinear responses, and can be used, with further improvements, to explore macroscopic quantum mechanics.

[1] National Laboratory for Physics Sciences at the Microscale, University of Science and Technology of China, Hefei 230026, China. [2] Department of Modern Physics, University of Science and Technology of China, Hefei 230026, China. [3] Synergetic Innovation Center of Quantum Information and Quantum Physics, University of Science and Technology of China, Hefei 230026, China. [4] High Magnetic Field Laboratory, Chinese Academy of Science, Hefei 230026, China. * These authors contributed equally to this work. Correspondence and requests for materials should be addressed to J.D. (email: djf@ustc.edu.cn).

Nonlinearity in micro- and nano-mechanical systems has been used to study processes such as fluctuation-enhanced dynamics[1–5], synchronization[6,7], mode mixing[8], noise control[9,10], signal amplification[11,12] and logic devices[13–15]. Its dynamics can be modelled by a driven Duffing oscillator equation as follows:

$$m\ddot{x} + \frac{m\omega_0}{Q}\dot{x} + kx + \alpha x^3 = F_{drive}\cos(\omega t). \quad (1)$$

Here $m$, $\omega_0$, $Q$ and $k = m\omega_0^2$ are the mass, resonance frequency, quality factor and linear spring constant, respectively. And $\alpha x^3$ is the Duffing nonlinearity with $\alpha$ the Duffing constant, that is, the cubic elastic constant. Under weak drive $F_{drive}$, the nonlinear response is negligible due to its cubic dependence on the amplitude $x$ of the resonator, and so the resonator behaves like a simple harmonic oscillator. This is the well-known Hooke's law of elasticity[16]. On the other hand, nontrivial dynamics of the resonator emerges when the drive is strong enough. A famous example is the occurrence of the driven Duffing biastability when the drive strength reaches a certain threshold $F_c$ (ref. 17), with the corresponding threshold power $P_c$.

Since $F_c$ is inversely proportional to $\sqrt{\alpha}$ as $F_c \sim \omega_0^3\sqrt{m^3/(\alpha Q^3)}$, the larger is $\alpha$, the weaker driving force is required for the system to reach nonlinear regime. One benefit of low driving force is low driving noise, while in those micro- and nano-mechanical systems reported in the literature, driving noise is far beyond the system's intrinsic Brownian thermal noise even at room temperature[1–5,9,11]. On the other hand, $P_c \sim m^2\omega_0^5/(\alpha Q^2)$, so increasing $\alpha$ can lower the threshold power, which is favourable in some scenarios[13–15]. In practice, since the system's fundamental parameters $m$, $\omega_0$ and $Q$ are limited by various factors, such as working bandwidth, material or fabrication, a universal way to increase and to tune $\alpha$ independently is significant.

What is more demanding is from quantum science, where strong nonlinearity can make quantum effects emerge from a classical harmonic resonator[18]. Such quantum nonlinearity is still elusive in macro-scale mechanical systems due to the naturally weak nonlinear response. Generally speaking, the emergence of quantum behaviour requires quantum nonlinear strength $g \propto \alpha x_{zpf}^4$ to overcome the system's decoherence[19–24], and since the quantum fluctuation $x_{zpf}$ is usually extremely tiny for a macro-scale resonator, generating ultra-strong nonlinear response $\alpha$ is of paramount importance.

In this work we demonstrate a system with a macroscopic mechanical resonator coupled to a single chemical bond, where the anharmonicity of the chemical bond deformation potential induces a giant nonlinear response of the resonator and can be tuned using external force. When driving to the nonlinear bi-states regime, stochastic transitions between bi-states are observed, which are demonstrated to be induced by the intrinsic Brownian thermal noise of the resonator.

## Results

**Theoretical model and density functional calculations.** Our system consists of a macroscopic mechanical resonator tightened to an anchor via a chemical bonding structure shown schematically in Fig. 1a. Strong nonlinearity is achieved when the resonator moves along $x$-direction by deforming the chemical bond. This is because, although the resonator alone follows the elasticity theory with linear dynamics, the response of the chemical bond is highly anharmonic. This is illustrated by the chemical bond's energy curve $U_{chem}(x)$ (Fig. 1b) obtained by density functional theory calculations (see Methods for details). For simplicity, $x = 0$ is set at the minimum of $U_{chem}(x)$ (Fig. 1b). By applying on the chemical bond an external control force $F$, the resonator's equilibrium position $x_{eq}$ can be tuned, and along with this the spring constant for sufficient small vibration is modified by $\Delta k = \partial^2 U_{chem}/\partial x^2$. As shown in Fig. 1c, when the resonator's equilibrium position is far away from the atom contact, the chemical bonding interaction is weak. By tuning the control force, that is, shifting the resonator's equilibrium position towards $x = 0$, the strength of $\Delta k$ first reaches a local maximum and then

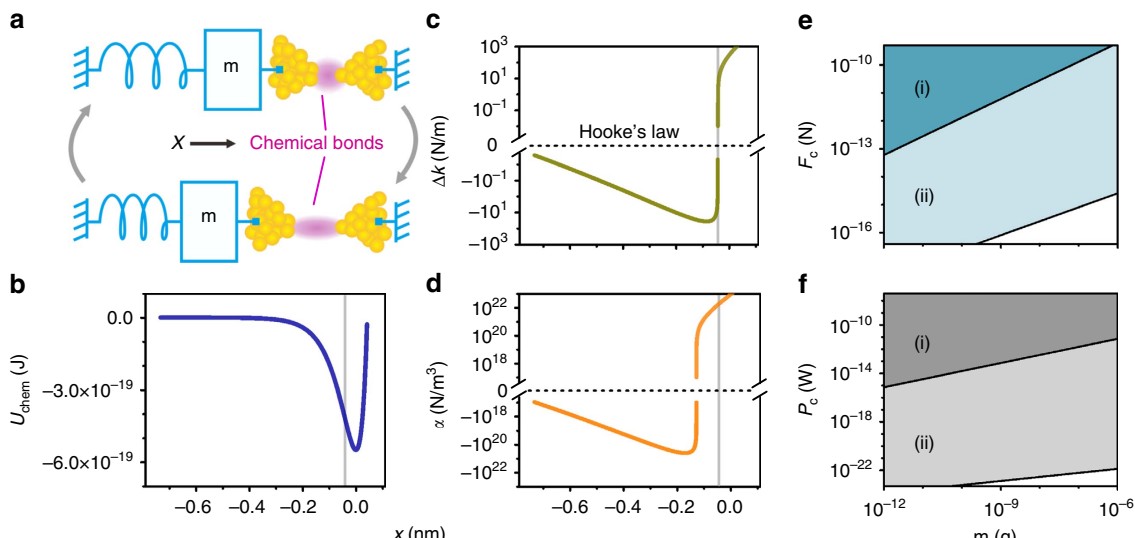

**Figure 1 | Concept of the system and theoretical results.** (**a**) A macroscopic resonator tightened to an anchor via chemical bonds. The displaced resonator can compress (top of panel) and stretch (bottom of panel) the chemical bond of gold-atom contact. (**b–d**) Density functional theory calculation results of (**b**) the chemical bonding interaction energy $U_{chem}(x)$, (**c**) the modified spring constant $\Delta k = \partial^2 U_{chem}/\partial x^2$ and (**d**) the enhanced nonlinearity coefficient $\alpha = (1/6)\partial^4 U_{chem}/\partial x^4$ as a function of the resonator displacement $x$. (**e,f**) Estimated threshold drive force $F_c$ and corresponding threshold power $P_c$ as a function of the resonator mass, $m$, with: (i) intrinsic nonlinear response of the resonator (dark blue, dark grey); and (ii) enhanced nonlinear response of the resonator (light blue, light grey) by chemical bonding interaction, at the point indicated as grey vertical line in **b–d**, where chemical bonding-induced linear response $\Delta k = 0$ (see Methods for details of the model).

drops to zero at the point where the chemical bonding attraction reaches maximum. The Duffing constant from chemical bonding, that is, $\alpha = (1/6)\partial^4 U_{chem}/\partial x^4$, changes differently (Fig. 1d): its strength first reaches a local maximum, then drops to zero before approaching the maximum attraction point, and finally changes its sign and increases in strength. At the maximum attraction point, $\alpha = 2 \times 10^{22} \, \text{N m}^{-3}$ is reached, while $\Delta k$ is zero.

Figure 1e,f shows the threshold driving force $F_c$ and the corresponding power $P_c$ as a function of the macroscopicity of the resonator (here characterized by the resonator's mass) with/without using the chemical bonding force-induced nonlinearity. With specific basic mechanical parameters, that is, resonance frequency $\omega_0$, quality factor $Q$ and mass $m$, introducing chemical bonding force can reduce $F_c$ as well as $P_c$ significantly. The term proportional to $x^3$ in the expansion of chemical bonding potential $U_{chem}$ and higher-order terms beyond the Duffing nonlinearity are also presented in our system, but are insignificant for bi-states dynamics studied in current experiments.

**Experimental realization of the system.** Now we demonstrate the above idea by using a macroscopic doubly clamped beam whose fundamental vibrational mode couples to a gold-atom contact, as shown in Fig. 2a. The atom contact is made by electric current migrations (see Methods) on a nano-bridge, which anchors the beam to a stiff electrode (Fig. 2b). The dimensions of beams is $l \times w \times t = 50 \times 1.5 \times 0.51 \, \mu m$. The mass of the beam is $\sim 0.2$ ng. The vibrations of the beam deform the gold–gold bonds of the contact along $x$ direction. The beam has a typical intrinsic frequency around $\omega_0/2\pi = 1.6$ MHz, with measured quality factor $Q$ ranging from 1,000 to over 3,000, depending on specific device. The device is placed in an ultrahigh vacuum chamber and has an environment temperature of 6 K.

We carried out the experiments (see Methods) with several devices. The measured beam's resonance frequency and the corresponding spring constant are plotted as functions of control

force $F$ in Fig. 3a–c for devices A to C correspondingly. Device A is in non-contact regime while B and C are in contact regime but with different atom structure in the contact. In typical contact regime devices, the frequency varies by about 1 MHz when the beam is pulled out to non-contact regime. Figure 3d–f shows the corresponding electron conductance results measured simultaneously as those for Fig. 3a–c. In the non-contact regime, the conductance due to quantum tunnelling shows a strong dependence on the control force and a value as low as $\sim 300 \, k\Omega^{-1}$ is reached, which confirms that a short-range chemical bonding force has been significantly involved[25], while in the contact regime, quantized conductance is observed and is insensitive to control force. Such a ballistic conductance is resulted from small number of metallic bonds[26].

From the dependence of the spring constant $k$ on the applied static force $F$, we can obtain the Duffing constant as

$$\alpha(F) = \frac{1}{6\xi^2} \left( \frac{\partial^2 k}{\partial F^2} k^2 + \left( \frac{\partial k}{\partial F} \right)^2 k \right), \qquad (2)$$

where $\xi$ is the shape factor with value the order of 1 and depends on the definition of the mode shape normalization of the device (Supplementary Note 1). To reliably obtain the Duffing constant from the data with noise, we have smoothed the $\alpha$ using a running average of five data points. Figure 3g–i plots the measured Duffing constant $\alpha$ as a function of control force $F$. In the single-bond case, the maximum strength of the Duffing constant achieved in our experiments is $(1.1 \pm 0.2) \times 10^{20} \, \text{N m}^{-3}$, with an enhancement of six orders relative to the estimated intrinsic nonlinearity due to the elongation of the beam $(\alpha_0 \approx 1 \times 10^{14} \, \text{N m}^{-3})$, and is many orders of magnitude larger in strength than those reported previously[11,12,27–31] (see Supplementary Table 1 for comparison to other systems). High tunability is easily reached by control force. In practical, both non-contact and contact regimes can be used depending on detailed applications. In the non-contact regime, jump-to-contact leads to system's instability (see Supplementary Note 1 for detail), while in contact regime such instability is naturally avoided. The frequency response of the device was studied by increasing the drive force, and the results are plotted in Fig. 3j–m, with Fig. 3j from device A, Fig. 3k,l from device B corresponding to regions I and II in Fig. 3h, respectively, and Fig. 3m from device C. The hysteresis responses are due to Duffing induced bi-states[17], with their directions agreeing with the signs of the nonlinear coefficients measured in Fig. 3g–i.

**Observation of thermally activated bi-states dynamics.** In the following we present a demonstration of a dynamical effect of the strong nonlinear response. Thermal fluctuation effects in nonlinear regime can lead to various complex dynamics, and have been studied in various systems, with an example being the centre of mass motion of optical trapped particles[32]. However, such a thermal nonlinear regime is exclusive in widely studied micro- and nano-mechanical systems. A well-known phenomenon is noise-activated bi-states transitions in a driven Duffing oscillator[33]. As a consequence of lacking strong enough nonlinear response, this phenomenon has only been previously observed by introducing strong artificial noise that far above the thermal fluctuation[1–5,9]. With the successful realization of strong nonlinear response here, we have observed in our system the bi-states transitions activated by the Brownian thermal noise even at cryogenic temperatures.

In the process, we chose a device (device D) of a relative high quality factor $Q = 3,100$, and tuned the control force to a point with the Duffing constant approximately $-1 \times 10^{17} \, \text{N m}^{-3}$. We avoided using the maximum absolute Duffing constant, in

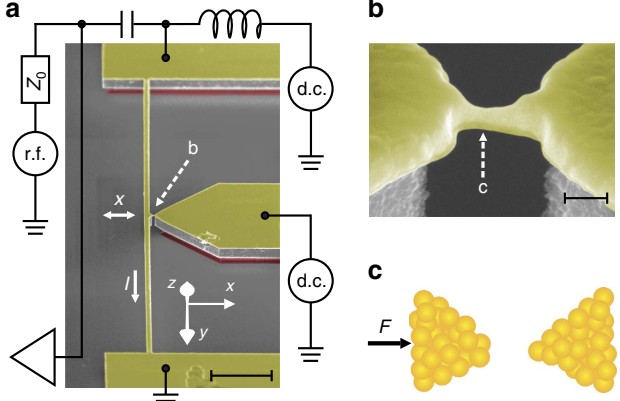

**Figure 2 | Experimental set-up.** (**a**) Scanning electron microscopy of a representative device in false colour. The macroscopic resonator is a doubly clamped silicon beam with thin layer of gold deposited on it, with dimension $l \times w \times t = 50 \times 1.5 \times 0.51 \, \mu m$ and total mass $\sim 0.2$ ng. The centre of the beam has horizontal displacement $x$. In the presence of a 6-T external magnetic field along the $z$ direction, the electric current ($I$) can excite and detect the motion of the beam, with the schematic circuits shown. Scale bar, 5 μm. (**b**) Nano-bridge connecting the beam to a stiff electrode, before experiments. Scale bar, 100 nm. (**c**) Cartoon plot of the atom contact generated on the nano-bridge indicated by 'c' in (**b**). The gold–gold bonding interaction is then tuned by force $F$, which is controlled by a d.c. current through the beam, and the electrostatic interaction of the contact is minimized applying a d.c. bias on the tip.

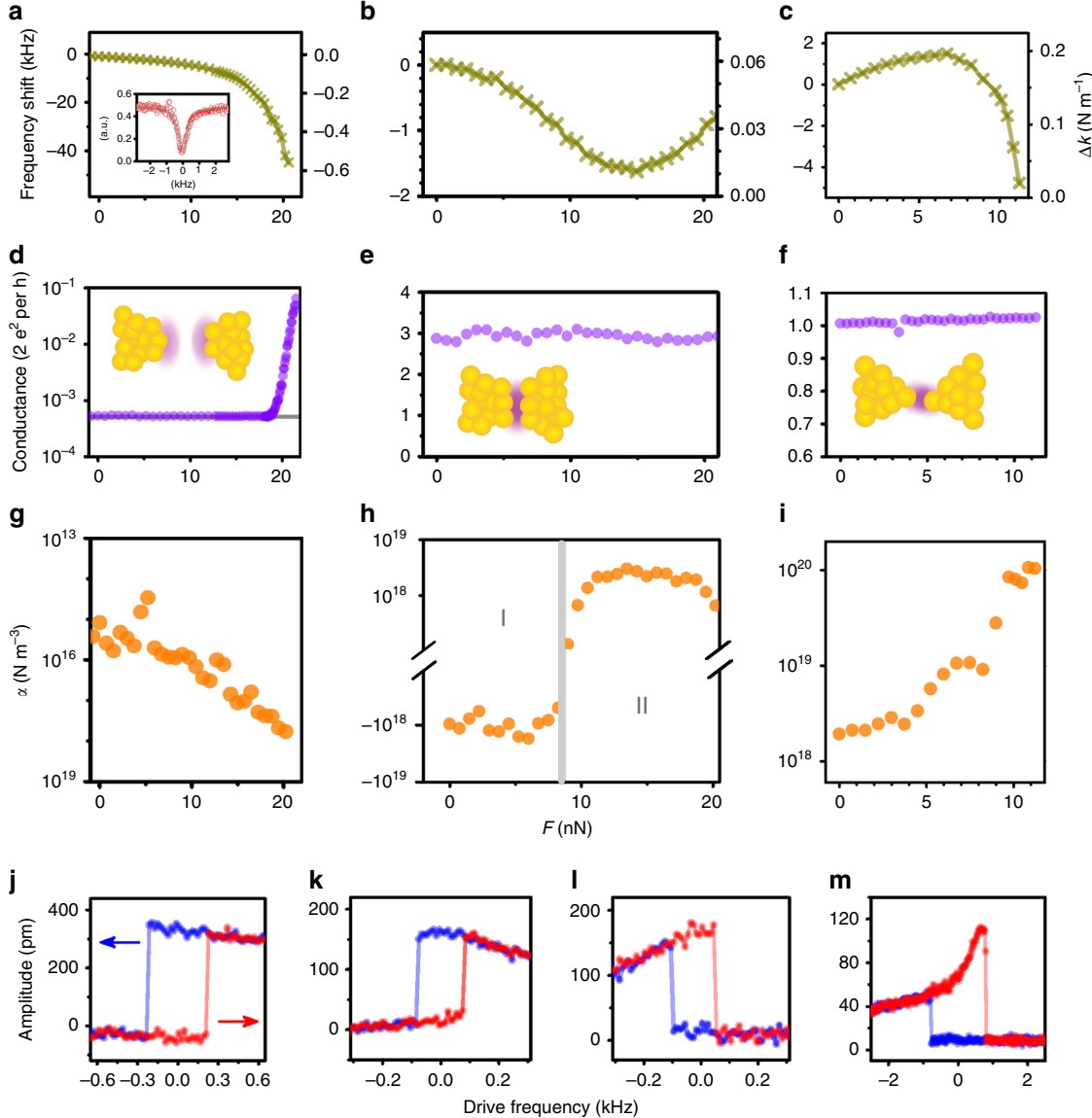

**Figure 3 | Tuning the nonlinear response by chemical bonding force.** (**a–c**) Frequency shift (left axis) and the corresponding effective spring constant change $\Delta k$ (right axis) as a function of the external control force $F$ for three different devices (A, B and C) which work in non-contact (device A) and contact regimes (devices B and C). Typical data to obtain the resonant frequency under weak drive with the beam in near linear regime are plotted in inset. (**d–f**) Conductance in unit of quantized conductance ($2e^2/h$) between atom contact measured simultaneously with **a–c** with inset cartoons the corresponding atom structures (schematic). The grey line in **d** indicates the noise level of measurement circuit. (**g–i**) Duffing constant $\alpha$ estimated from **a–c** correspondingly. (**j–m**) Typical hysteresis response under drive frequency sweeping corresponding to device A (**j**) device B ((**k**) for region I and (**l**) for region II) and device C (**m**). Note that the frequency shift, effective string constant change ($\Delta k$), control force $F$ and driving frequency are all relative with large constants being subtracted for the ease of displaying.

which case the corresponding bi-states amplitude would be too small and that would make the motion sensing the noise. We drove the system to the bistable state regime and observed the switching behaviour by recording the amplitude of vibrations of the beam, as shown by the green trajectory in Fig. 4a.

To verify that such switchings are indeed from the Brownian thermal noise of the beam, we introduced an amplitude modulation to the driving signal of the form $F_{drive}(t) = F_{drive} + \delta F_{drive} \cos(\Omega t)$, with $\Omega = 0.5\,Hz$ and $\delta F_{drive} = 0.18\,pN$. The amplitudes of the vibration of the beam show periodic switchings (Fig. 4a, purple trajectory) instead of random switchings. Figure 4b shows the corresponding power density, $S_{mod}(\Omega)$ with modulations and $S_{noise}(\Omega)$ without modulations, from which the signal to noise ratio (SNR) is calculated with

$SNR(\Omega) = S_{mod}(\Omega)/S_{noise}(\Omega)$. By using the standard stochastic resonance theory (Methods), we estimated the total force noise as $\sqrt{S_{total}^F} = (3.6 \pm 0.6) \times 10^{-16}\,N\,Hz^{-1/2}$. It agrees nicely with the resonator's Brownian thermal noise, estimated[34] by $\sqrt{S_{th}^F} = 4m\omega_0 k_B T/Q$ as $3.3 \times 10^{-16}\,N\,Hz^{-1/2}$. In the limit of $\Omega$ approaching zero, the modulation becomes a perturbative force signal $\delta F_{drive} \mathrm{e} \cos(\omega t)$ added to the driving force with the same phase. In the rotating frame of the driving signal, the bi-states dynamics is described by an over-damped double well whose shape is tuned by $\delta F_{drive}$ (ref. 33), and becomes sensitive to the tuning near the bifurcation point. By applying a weak force $\delta F_{drive} = 2\,fN$ that is only several times larger than the resonator's

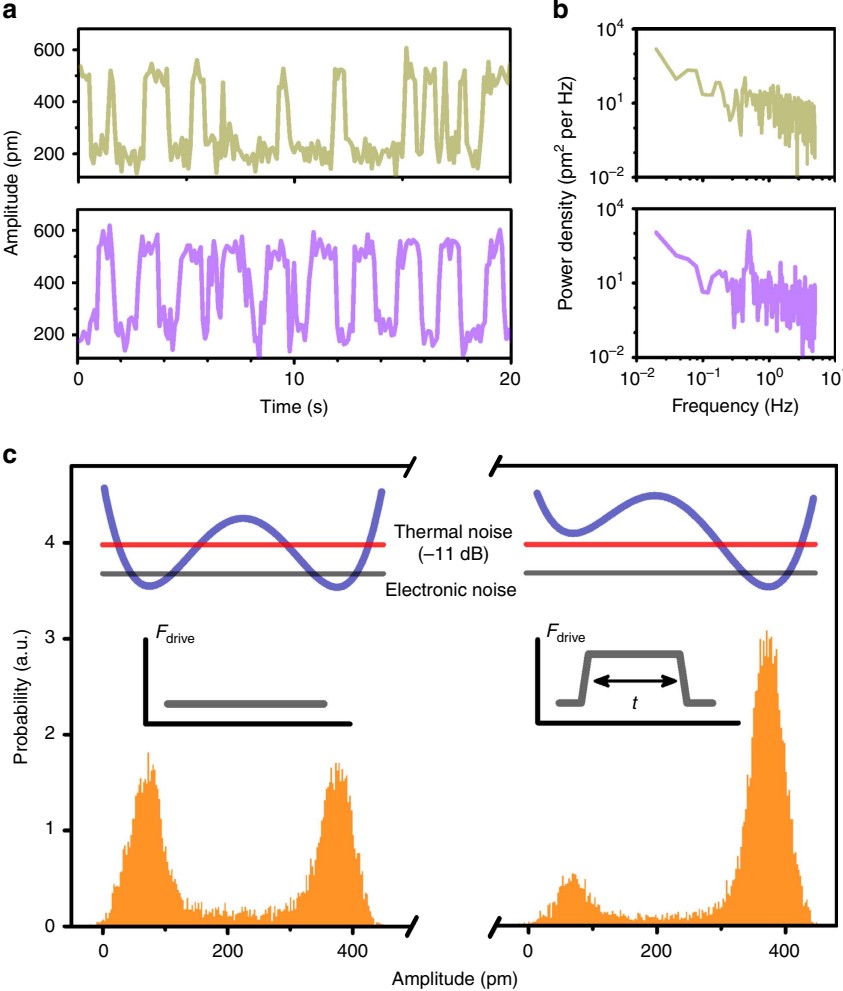

**Figure 4 | Brownian thermal noise induced bi-states transitions.** (**a**) Switching between bistable states with constant-amplitude drive (upper panel) and modulated-amplitude drive (lower panel). To observe bi-states transitions, the driving frequency is fixed in the middle point of hysteresis response and the drive amplitude is $F_{drive} = 4$ pN, and the modulation frequency and amplitude are $\Omega = 0.5$ Hz and $\delta F_{drive} = 0.18$ pN, respectively. (**b**) Power spectrum density. Compared with the constant-amplitude drive (upper panel), the amplitude-modulated drive (lower panel) induces an additional peak in the power spectrum density at the modulation frequency $\Omega = 0.5$ Hz. (**c**) The amplitude distribution of the bistable resonator depends on the driving amplitude. Very different histograms for constant drive with $F_{drive} = 4.000$ pN (left panel) and $F_{drive} = 4.002$ pN (right panel) are observed. The change in amplitude distribution can be understood using an effective double-well potential (sketched in the inset). The total electronic noise is 11 dB below the Brownian thermal noise in our experiment.

Brownian thermal noise force, significant changes in the statistic distribution are observed, as shown in Fig. 4c. The sensitivity of such a response to the external force is limited by the total force noise $\sqrt{S_{total}^F}$. It is noted that, in our device, in addition to the thermal noise, the electronic noise from the circuit also contributes to the total force noise, but with a weak power density of $S_{elec}^F = S_{total}^F - S_{th}^F$. We estimated that $S_{elec}^F$ is 11 dB below the $S_{th}^F$ level, corresponding to an electronic noise temperature of 500 mK, and is mainly limited by our room-temperature detection circuit (see Supplementary Table 2 for detailed data).

## Discussion

In conclusion, we have demonstrated a highly controllable nonlinearity in a macroscopic mechanical system, with its fluctuation dynamics dominated by Brownian thermal noise. The universal existence and the small scale of the chemical bonding force make our method applicable to current widely used micro- and nano-mechanical systems in improving their nonlinear responses. With further improvements, nonlinearity-induced quantum behaviours in macroscopic mechanics[19–24] are foreseeable in the type of systems described here.

## Methods

**Theoretical description of the system.** Density functional theory calculations were used to estimate qualitatively the chemical bonding force (see Supplementary Note 1 for detailed description). Owing to stability considerations, a small structure of two gold-atom clusters was adopted in the calculation (see Supplementary Fig. 1 for data). By changing only the relative position $x$ of the left cluster relative to the right one while otherwise keeping the relative positions of all the gold atoms fixed, we obtained a chemical bonding energy function $U_{chem}(x)$, from which the nonlinearity is calculated as the fourth-order derivative. Other structures of clusters are also considered (see Supplementary Fig. 2 for data). Owing to the existing systematic error in performing density functional theory calculations, the second-order derivative of the total energy shows slightly an oscillatory behaviour at large distance range. We smoothed the calculated data by fitting it to the function $\frac{A}{x^a} + \frac{B}{x^b}$. In reality, however, the relaxation of gold atoms can occur, which may modify the energy–displacement curve $U_{chem}(x)$. We have considered this effect and find out that the results do not change significantly. To estimate the long-range forces, we considered analytically the van der Waals attraction and the electrostatic force[25] for the geometric model of two gold cylinders close to each other, similar to the case of our nano-bridge structure and in modelling nonlinear

dynamics, we have dropped the quadratic nonlinearity in the chemical interaction (see Supplementary Fig. 3 for data). To estimate the intrinsic macroscopic mechanical property, we modelled the macroscopic resonator as a doubly clamped beam. The lowest vibrational mode has a weak intrinsic nonlinearity due to elongation. In calculating the threshold drive force $F_c$ and the corresponding power $P_c$ in Fig. 1e,f, we took the mechanical quality factor 3,000 and the ratio between the beam's thickness and length $t/l \geq 0.005$. We also estimated the electrostatic force-induced nonlinear response based on the same doubly clamped beam (see Supplementary Fig. 4 for data).

**Fabrication of the device.** The device was fabricated on a commercial silicon-on-insulator wafer using nanolithography (see Supplementary Note 2 for detailed descriptions). The nano-bridge connecting the beam and the stiff electrode is about 200 nm in length and 80 nm in diameter. Focus-ion beam was adopted to narrow it down to < 50 nm. Once such a device, that is, the doubly clamped beam with a suspended nano-bridge, was successfully fabricated, it was placed in the ultrahigh vacuum environment of the cryogenic system, and then the bridge was electro-migrated to form an atomic point contact[35]. After the atom contact was produced, the beam was disconnected from the stiff electrode and its lowest vibrational mode was measured. The position of the equilibrium of the beam is controlled by applying a d.c. current through the beam so that the atom interaction can be tuned.

**Device characterization and electronic noise.** The resonant frequency $(2\pi)^{-1}\omega$ of the device under a control force $F$ is measured directly when the device is working in the linear regime under a weak drive. The spring constant $k$ is calculated as $k = m\omega^2$. Here the effective mass $m$ for the first vibrational mode of the beam is estimated by using finite element simulation, which is widely used for similar systems[36]. The electric conductance is measured by using $G = dI(dV)^{-1}$.

The Brownian thermal noise on the beam is calculated to be $3.3 \times 10^{-16}$ N Hz$^{-1/2}$. On the basis of this and the calculated mass, the measured resonator frequency and quality factor, we estimate the thermal motion amplitude noise on resonance as $1.0 \times 10^{-13}$ m Hz$^{-1/2}$ (refs 34,36), which cannot be measured directly due to the sensitivity limitation of our room-temperature measurement electrics. We modelled the mechanical resonator as LCR elements following standard procedure[37] (see Supplementary Note 3 for detailed descriptions), and analysed the noise in the equivalence electric circuit (see Supplementary Fig. 5 for circuit diagram). The separation of the atom contact can be controlled with a precision of about 1 pm without any feedback control for thousands of seconds (see Supplementary Fig. 6 for the data), and once jump-to-contact occurred, a strong enough control force is used to pulled the tip out again and device's main nonlinear character is commonly preserved (see Supplementary Fig. 7 for data). The intrinsic nonlinearity is measured from the frequency response to the driving strength in a standard way based on the Duffing nonlinear model[17] (see Supplementary Fig. 8 for data). To do this, we have pulled the beam far apart from the stiff electrode with contact separation larger than 30 nm, so the interaction due to the atom contact is negligible. A 6-T magnetic field is used in our experiments so that control force can be generated using a small current. This decreases the system total quality factor from its intrinsic value (see Supplementary Fig. 9 for data). We also applied a voltage bias < 200 mV between the beam and the stiff electrode to compensate the contact potential of the atom contact so that the contribution of electrostatic force in our experiments is minimized (see Supplementary Fig. 10 for data).

In our experiment, there are mainly three sources of electronic noise. The first one is the Johnson–Nyquist noise $S_R^V = 4Rk_BT$, with $T$ room temperature and $R$ the circuit's resistant, which produces a force noise with power density $S_R^F$. The second one is the current leakage from the input of the voltage preamplifier $S_{ba}^I$. We model the preamplifier back-action by a current source similar to that described in ref. 38. In doing this, we assume that there is no correlation between voltage imprecision $S_{im}^V$ and the back-action $S_{ba}^I$ (ref 39), so $S_{ba}^I$ leads to a force noise with power density $S_{ba}^F$. Another one is from the radio frequency driving signal, which generates a phase noise and works as an equivalent force noise on the beam with power density $S_{pha}^F$. We estimate the total electronic-induced force noise as $S_{elec}^F = S_R^F + S_{ba}^F + S_{pha}^F$, and in the experiments, $\sqrt{S_{elec}^F} = 0.9 \times 10^{-16}$ N Hz$^{-1/2}$, with power density about 11 dB below the $S_{th}^F$.

**Measurement of the total force noise on the beam.** The dynamics of our system is modelled by the standard Duffing oscillator of equation of motion

$$m\ddot{x} + \gamma\dot{x} + m\omega_0^2 x + \alpha x^3 = F_{\text{drive}}(t)\cos(\omega t) + F_{\text{noise}}(t). \quad (3)$$

where the dissipation rate $\gamma = m\omega_0/Q$, $F_{\text{drive}}(t) = F_{\text{drive}} + \delta F_{\text{drive}}\cos(\Omega t)$ is the amplitude-modulated driving force and $F_{\text{noise}}(t)$ is the total noise with power density $S_{\text{total}}^F$. The minimum force that drives the system into the nonlinear regime where bifurcation occurs is $F_c$, with the corresponding vibration frequency being $(2\pi)^{-1}\omega_c$, and the amplitude $x_c$ can be calculated from equation (3)[17]. Near the nonlinear bifurcation point, we transform this equation to an over-damped one by following the standard procedure[33]. For the case of modulation frequency $\Omega$ being much smaller than the decay rate $\omega_0/Q$ of the system, the SNR is related to the

system's noise power as[40]

$$\text{SNR} = \pi \frac{\gamma_k}{\delta\omega} x_m^2 \left(\frac{\delta F_{\text{drive}} m\omega}{S_{\text{total}}^F}\right)^2, \quad (4)$$

with $x_m$ the half of the vibration amplitude of the bistable states, $\delta\omega$ the nonlinearity frequency-induced shift from linear resonance peak and $\gamma_k$ the measured random switching rate without modulation. So from the measured SNR we obtained $S_{\text{total}}^F$.

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

## Acknowledgements

We thank L. Jiang for many stimulating discussions and comments and for his help in improving the manuscript. We thank Y.X. Liu, Z.Q. Yin, L. Tian C.P. Sun, and M. Lukin for helpful discussions. We thank Yiqun Wang from Suzhou Institute of Nano-Tech and Nano-Bionics for nano-fabricating supports and Guizhou Provincial Key laboratory of Computational Nano-Material Science for calculation supports. This work was supported by the 973 Program (Grant No. 2013CB921800), the NNSFC (Grant Nos. 11227901, 91021005, 11104262, 31470835, 21233007, 21303175, 21322305, 11374305 and 11274299), and the 'Strategic Priority Research Program (B)' of the CAS (Grant Nos. XDB01030400 and 01020000).

## Author contributions

J.D. supervised the experiments; J.D. and P.H. proposed the idea and designed the experimental proposal; P.H., J.Z., L.Z., W.D. and F.X. prepared the experimental set-up; P.H., J.Z., L.Z., S.L. and C.J. performed the experiments; J.Z. fabricated the device; D.H. and X.Z. carried out the theoretical calculation; M.C. carried out the finite element simulation; P.H., C.D., F.X. and J.D. wrote the paper; all authors analysed the data, discussed the results and commented on the manuscript.

## Additional information

**Competing financial interests:** The authors declare no competing financial interests.

