## [Peer review file · Nature Communications]

Transferred manuscripts:

Replies to Referees' reports

Firstly, we would like to thank the referees for their hard work in reviewing our work. We have carefully analyzed all the issues raised by them, checked our calculations and data analyses again and made necessary revisions to the manuscript accordingly. Point to point replies (Q & A) and a list of changes are given below.

Referee1

The authors addressed many of my comments but there are still a few things that I don't understand and a few typos in the paper.

Q1. The new Figure 3A presents more data but still doesn't make it clear for me to understand. Perhaps I am missing something obvious, but I don't understand how the authors measure the frequency shift and extract the change in spring constant and the duffing parameter alpha. I understand the equations but I don't understand how the experimental results lead to the plots. For example, in Figure 3b and 3c how does a frequency shift = 0 correspond to a spring constant change = 0.06. In the non-contact regime it is 0. Is f_0 still defined the same way? If f_0 is the intrinsic frequency without the gold contact where does the delta k come from when frequency shift = 0?

A1:

The frequency is measured directly, the change of spring constant is calculated from the frequency shift (said in Methods), and then the duffing parameter alpha is calculated using equation (2) in the maintext.

In our plots for Fig 3b and 3c, the frequency shift and change of spring constant were relative values (with an arbitrary chosen large constants subtracted) and that caused the misunderstanding. So we update the figures and captions.

We have revised the 2nd paragraph of the section 'experimental realization of the system' and 'device characterization and noise' (in Methods) to make all those clear (List of changes, 4)

Q2:

Figure 3j -3m are not discussed in the main text. I don't understand the origin of the hysteresis response in Figure 3j.

A2:

We have added to the main text “the frequency response of the device was studied by increasing the drive force, and the results are plotted in Fig.~3j to 3m,... The hysteresis responses are due to Duffing induced bi-states, with their directions agreeing with the signs of the nonlinear coefficients measured in Fig.~3g to 3i” (List of changes, 4).

Q3:

There are also a few typos:

1. Page 4: Electric transportation should be electron transport or something like this
2. Ballistic is ballistic
3. Page 4: Figure 3g - 3i plots alpha as function of control force.
4. Page 5: Figure 3b should say Figure 4b
5. Figure S1 caption title: Nonlinearity
6. Figure S3 caption title "quadratic"
7. Figure S3 caption "frequency" and "quadratic"
8. Figure S7 caption: "frequency"

A3:

We thank Referee 1 and have checked the manuscript for typos, grammars and terminologies.

Referee2

I appreciate the effort that the authors put in answering my (and the other reviewer's) comments. However, I disagree that my points have been met, maybe because some of them were not understood or because I do not understand their answer. So I am against publication of this paper as it is.

Q4:

- In the abstract it might be good if you remove "applications". If you call on "fundamental studies" then that is great. And there is a whole lot of things that you can do when thermomechanical noise can make your system enter into the NL regime. However, for applications, this is not going to work. Unfortunately, going nonlinear (which indeed helps for some points) deteriorates performance. Please keep in mind that I said that it can be good for fundamental studies, but not for applications. there is no actual application where devices operate in their nonlinear regime. The only place I can think of that this has been applied and it's beneficial is in synchronization. But there you might want to have a very well controllable and predictable nonlinear coefficient, which I don't think it's the case here.

A4:

(1) Regarding to the applications of nonlinearity in macroscopic mechanic system, by ‘application’, we have followed the common practice in many research articles by including those that are still under study, rather than just those being commercialized.

In this respect, we are afraid we could not agree with Referee 2 by saying “...there is no actual application where devices operate in their nonlinear regime...”. Apart from synchronization mentioned by Referee 2, information storage (as shown in Ref 13, 14 and 15) is another widely-studied example, which requires the system working in NL regime, with increasing nonlinear response being advantageous (see the last paragraph of Ref 13).

We understand that in some cases, strong nonlinear response is not favorable but those are not relevant to our study that focuses on generating giant and tunable nonlinearity.

(2) Regarding to “but there you might want to have a very well controllable and predictable nonlinear coefficient, which I don't think it's the case here”, we would like to emphasis that the nonlinear coefficient of our devices are tunable by about six orders of magnitudes, and the chip-based structure has very high stability as shown in SI Fig S6. Furthermore, to our knowledge, such a controllability is much better than reported previously.

We revised the first introductory sentence in abstract as “Nonlinearity in macroscopic mechanical systems will lead to abundant phenomena for fundamental studies and potential innovative applications” (List of changes, 1)

Q5:

- The characterization of the nonlinearity is incomplete and flawed. As I said last time, alpha is not univocally defined. It depends on the definition of the mode shape normalization that alpha will take one value or another. That is why talking in general about the value of alpha without defining the normalization explicitly, it is not correct.

A5:

We have done the characterization of nonlinearity carefully in our work.

The normalization process has been performed by considering the shape of the beam’s vibration mode by adopting the commonly-used method in Ref. S8 (detailed descriptions in section S1.4 of SI). We have modified the expressions regarding to the shape factor in the text to make this clearer (List of changes, 6).

Q6:

- The measurements for alpha they present in the SI require two things: that the transduction is linear and that the electrical background is negligible. Without

commenting on the former point, the latter is clearly not met - as we can see in Fig. SI8.

A6:

Referee 2's concern is reasonable in some cases. For our case here, we have explicitly included the electronic background in our measurement circuit, as described in SI section S2.2 and S2.3, so the vibration of beam can be estimated from the electric response with background. We then follow the standard method (ref. S9) to obtain the intrinsic nonlinear response of the free beam from frequency response, and the results agree well with theoretical estimation based on the elasticity of doubly clamped beam structure. So the value measured is solid.

Q7:

- I do not fully understand the noise analysis. what I think of is thermomechanical noise, then johnson's noise in the resistor, then amplifier noise (BOTH current and voltage noise). why the latter is not taken into account escapes me. Also, the way they refer to the current noise coming from the amplifier as a back-action noise is a bit weird for me as well. Finally, they claim in their response that in the SI they do a noise analysis comparing all the noise sources but I don't see all, as I am missing the thermomechanical noise. Maybe I did not get the full file.

A7:

The amplifier noise was carefully measured and clearly described in S3.1 of SI, the procedure of modeling the amplifier noise has followed the standard methods (reference 39 and S12) , the voltage noise is measurement imprecision and does not contribute to the system's total force noise (see, for example, Ref. 39 for a clear description of this). The comparison between different noises has been included in the original version of manuscript and now in revised version as a table (table S2 in SI) for direct comparison.

Q8:

- Square nonlinearity - a similar situation comes to β . they say in their response that they have analyzed and seen that the contribution of β is very small, but in the SI it is not very clear. I would like to have a clear estimation of β and α and I would like the authors to tell me how much the response changes when using one or the other. This is important mostly for the part where they estimate " α ". if they have " β " then the result will be different. it is evident, has been proven, that you can project the contribution of β into a cubic term nonlinearity. but if the square term exists, its effect might be apparent before the one of the cubic nonlinearity. The authors actually present Fig. S3A which I believe has a wrong legend or caption. if I take for granted that the values they give for β are the ones they calculate (the same way they calculate α , they should be able to calculate β) then I can see that if the amplitude is 1pm, the term with β is going to be 2 orders of magnitude larger than the term with α . For them to be the same, the amplitude should be 1

Angstrom, 0.1 nm. And I doubt the authors have these amplitudes when they are in the strong NL regime.

A8:

Firstly, typos in Fig. S3 have been corrected.

We have checked our simulation results presented in Fig. S3 and the results are correct.

We note that, in our work we concentrate on the bi-states dynamics caused by cubic nonlinearity in the frequency domain near the resonant point of the first vibration mode. Considering an oscillation $x \sim \cos(\omega t)$, the squared term βx^2 contributes, to the first order of perturbation, to the frequency components of d.c. and 2ω and therefore does not change directly the characteristic of bi-state dynamics.

In this revision we have reworded the related sentences to avoid possible misunderstanding (List of changes, 3).

Q9:

- that the authors have higher orders of nonlinearity is evident from their own data. One cannot show Fig. 3.m and Fig. 3.k and then say that they only have alpha. It is just weird.

A9:

In our work, we would like to emphasize that we focus on generating and the characterization of cubic nonlinearity.

We know that it is possible to have high-order nonlinearity contributions in our systems, especially in the new data (Fig. 3k-m) presented from the 2nd version of our manuscript, where stronger nonlinearities are observed. However, such high-order nonlinearities have no significant role in our study of bi-states dynamics presented in figure 4, where a modest strength of cubic nonlinearity (corresponding to figure 3j, in non-contact regime) has been used.

In this revision we have reworded the related sentences to avoid possible misunderstanding (List of changes, 3).

Q10:

- In the caption for Fig 4 - the authors cite $F=4.000\text{pN}$ and 4.002 pN . how can they apply this 2 fN force difference? can you show me the numbers?

A10:

The calibration of force is based on Lorentz force by following the standard procedure (Ref. 36 for example) and is described in SI section S2.3. The strength of the force is proportional to the amplitude of RF signal, which is from a commercial RF generator with 16 bit amplitude precision [the order of 10^{-5}]. The force difference of around 2 fN is reliable.

Q11:

- Coming back to my original review - I asked that why don't the authors calibrate their motion using thermomechanical noise and I still do not understand the answer. This is the most typical way of calibrating the amplitude and, at least, they should be comparing to their estimation to make sure it's the same. unless they cannot measure thermomechanical noise, but the authors claim they can.

A11:

We checked previous version of our manuscript carefully, and are certain that we have made no claim on direct measuring thermomechanical noise.

Instead of calibrating the motion by measuring thermal motion directly, we adopted the method based on magneto-drive, which is also widely used (see Ref. 36 for example).

In our experiments, we measured the bi-states dynamics, from which the thermomechanical noise was determined.

Q12

- The discussion on power consumption is a bit confusing also. When you, with magnetomotive, make a force on your beam you require the pass of a current through a resistor. this, evidently is going to dissipate power (joule effect - leads to heating of the beam). If I apply this force either using capacitive//electrostatic or piezoelectric actuation, then virtually no power is dissipated as they are non-dissipative actuation mechanisms. If the authors meant that they can drive the system into the NL regime applying a lower power output from their network analyzer, that is a different story. but still that would depend on the impedance matching, etc.

A12:

The power consumption concerned in our work is the intrinsic property of the beam and is clear defined. It has nothing to do with the specific external measurement networks, which is not our concern.

List of changes

1. Abstract:
The first sentence is changed into “Nonlinearity in macroscopic mechanical systems will lead to abundant phenomena for fundamental studies and potential innovative applications.”
2. Page 1, in the two paragraphs below Eq. (1):
The quantity P_c is renamed as ‘threshold power’ for clarity.
3. The end of the first paragraph of page 4:
We add “higher-order terms beyond the Duffing nonlinearity can also be present in our system, but are insignificant for bi-states dynamics studied in current experiments.”
4. About Fig. 3:
 - (1) Page 5, the end of paragraph below Eq. (2): two sentences about Fig. 3j to 3m and ‘the hysteresis’ are added
 - (2) Page 8, first paragraph of the section ‘Device characterization and electronic noise’: A whole paragraph is added on obtaining the quantities plotted in Fig.3
 - (3) Page 17, the end of the caption of figure 3: added “In the plot, the frequency shift, the effective string constant change (Δk), the control force F and the driving frequency are all relative values with large constants being subtracted for the ease of displaying.”
5. SI Section 1.4, below Eq. (S8):
More detailed description about the mode normalization is provided.
6. Miscellaneous minor corrections to typos, grammars, expressions and formats

Reviewers' Comments:

Reviewer #1 (Remarks to the Author)

The authors sufficiently addressed my concerns. I think they can do a better job addressing referee 2 with the below edits.

1. I think the disagreement about "applications" is a matter of style. I think both the reviewer and the author make legitimate points and I leave it up to the authors to decide what to call applications. I don't think they have to limit the word "application" to something already commercialized. If the authors want to be extra cautious, they could just say "potential applications".
2. Q7: The authors cite <http://www.nature.com/nature/journal/v406/n6799/full/4061039a0.html> which is where I assume they got the term back-action to refer to the current noise. To avoid confusion, they might consider adding a sentence that they call this back-action following the notation of reference 13 in supplementary.
3. Q11: In Methods on page 11 the authors should say "The Brownian thermal noise of the beam is calculated to be $3.3 \times 10^{-16} \text{ N} \cdot \text{Hz}^{-1/2}$ based on the measured resonator frequency and quality factor, and calculated mass. They can then say that they use this estimate to make a further estimate of the thermal noise amplitude. They can also add a sentence that says they are unable to measure the thermal noise amplitude for such and such a reason. This will avoid confusion about whether the authors measure. They estimate the thermal noise from the measured quality factor, resonant frequency, and estimated mass but the sentence does mislead the reader into thinking it was measured.

List of changes

We would like to thank referee 1's remarks and have made necessary revisions to the manuscript accordingly. We also have checked the editorial requests and have made necessary revisions. Point to point replies (Q & A) and a list of changes are given below.

Referee1

1. I think the disagreement about "applications" is a matter of style. I think both the reviewer and the author make legitimate points and I leave it up to the authors to decide what to call applications. I don't think they have to limit the word "application" to something already commercialized. If the authors want to be extra cautious, they could just say "potential applications".

Reply:

We thank Reviewer's support and suggestion. The phrase "potential applications" is therefore kept in our abstract.

2. Q7: The authors cite <http://www.nature.com/nature/journal/v406/n6799/full/4061039a0.html> which is where I assume they got the term back-action to refer to the current noise. To avoid confusion, they might consider adding a sentence that they call this back-action following the notation of reference 13 in supplementary.

Reply:

We have added a sentence following the suggestion, see list of change 1.

3. Q11: In Methods on page 11 the authors should say "The Brownian thermal noise of the beam is calculated to be $3.3 \times 10^{-16} \text{ N} \cdot \text{Hz}^{-1/2}$ based on the measured resonator frequency and quality factor, and calculated mass. They can then say that they use this estimate to make a further estimate of the thermal noise amplitude. They can also add a sentence that says they are unable to measure the thermal noise amplitude for such and such a reason. This will avoid confusion about whether the authors measure. They estimate the thermal noise from the measured quality factor, resonant frequency, and estimated mass but the sentence does mislead the reader into thinking it was measured.

Reply:

We have added a sentence following the suggestions see list of change 1.

List of changes

1. By following Referee 1's suggestion, in Paragraph 2 of Page 11, two sentences starting with "which can not be measured ..." are revised, and in Line 3, Page 19

in SI: The sentence starting with “The second is ..” is revised.

2. In page 3 of main text: a paragraph summarizing the main results is added at the end of the introduction section by following the policies and format requirements.
3. The format titles, figures etc. of SI have been rearranged by following the policies and format requirements.